# 3D Digital Preservation, Presentation, and Interpretation of Wooden Cultural Heritage on the Example of Sculptures of the FormaViva Kostanjevica Na Krki Collection

Andrej Učakar [1], Ana Sterle [2], Martina Vuga [2], Tamara Trček Pečak [2], Denis Trček [3], Jure Ahtik [1], Karin Košak [4], Deja Muck [1], Helena Gabrijelčič Tomc [1,*] and Tanja Nuša Kočevar [4]

[1] Faculty of Natural Sciences and Engineering, Department of Textiles, Graphic Arts and Design, Chair of Information and Graphic Arts Technology, University of Ljubljana, Snežniška 5, 1000 Ljubljana, Slovenia

[2] Department of Conservation and Restoration, Academy of Fine Arts and Design, University of Ljubljana, Erjavčeva ulica 23, 1000 Ljubljana, Slovenia

[3] Faculty of Computer and Information Science, University of Ljubljana, Večna pot 113, 1000 Ljubljana, Slovenia

[4] Faculty of Natural Sciences and Engineering, Department of Textiles, Graphic Arts and Design, Chair of Textile and Fashion Design, University of Ljubljana, Snežniška 5, 1000 Ljubljana, Slovenia

* Correspondence: helena.gabrijelcictomc@ntf.uni-lj.si

**Abstract:** The paper presents an interdisciplinary approach to the treatment of the FormaViva collection of wooden sculptures exhibited outdoors in a natural environment near the Božidar Jakac Art Museum in Kostanjevica na Krki in Slovenia. The study focuses on 3D graphic representations of sculptures created with photogrammetry and 3D modelling. The results are photorealistic renderings, interactive presentations, 3D printed reproductions, jewellery, and interpretive animations. The research results show that graphic documentation techniques on 3D models allow for a more detailed investigation of the original structural identity of the sculpture. By incorporating 3D and interactive technologies, we are expanding the usability of cultural heritage objects. By using interpretive techniques that have led to jewellery and interpretive animations in our research, we not only breathe new life into the sculptures, but also enrich the stories of the sculptures with our own experiences of the sculptural work.

**Keywords:** 3D modelling; 3D printing; animation; cultural heritage; jewellery; photogrammetry; preservation; representation; wooden sculptures

## 1. Introduction

In recent years, the rapid developments of digitization have reached cultural heritage. Technologies such as digital archiving, documentation, and reconstruction, 360° capturing, 3D laser scanning, modelling, and printing are influencing the way cultural heritage is maintained, managed, and preserved, as well as the way it is observed by visitors and audiences in general. In addition to the technologies, augmented, mixed, and virtual reality are also increasingly being used in the cultural heritage sector [1].

These new technologies can help increase the reach of cultural heritage. In some cases, they can extend and expand the life of cultural heritage when its original physical manifestation is no longer present [1].

Technologies are now financially affordable, and quality is increasing. The decline in price [2] and the increasing accuracy of 3D printing technologies [3] are now providing new ways to represent and analyse cultural heritage. This "digital cultural heritage", also referred to as "heritage information resource" [4], can extend the life of heritage and become accessible to everyone.

A cultural heritage is defined by its historical relevance and artistic qualities, but also by its cultural values, identity, associations, and ability to evoke memories [5].

If we focus on 3D printing of sculptures, this opens new possibilities in the field of research and accessibility of this cultural heritage.

One of the most important and probably most obvious advantages of 3D printing is its high flexibility compared to traditional methods of producing copies of sculptures. Three-dimensional-printed prototypes can be touched without fear of damaging them. Thus, they represent a new way of interacting with exhibits. There are many factors to consider if we would like to get as close as possible to the original. For example, we also need to consider texture, weight, and final appearance, which is sometimes difficult to achieve with 3D printing [6,7] and "material poetics" may be lost [8].

Three-dimensional capturing technologies and processing methods differ in terms of accuracy and thus the final results of 3D printing. When we capture with different technologies, we obtain a point cloud that depends on the accuracy of the capture. During processing, different filters are used to optimize the digital model, which in turn can affect the geometry of the model. Metrics obtained by photogrammetry or laser scanning often contain a lot of noise [3,6].

We can also create 3D prototypes of the exhibits using the 3D modelling process, but we must be aware that such models are ultimately only an approximation of the original, as they are the result of interpretation [6].

Each artwork requires a unique approach, especially in the capture of the artwork's digital data and data processing. Depending on the characteristics of the artwork, such as size, construction, material, age, and condition, we determine the conditions for capturing, equipment and further data processing.

For the capture of wooden sculptures, both photogrammetry and scanning prove to be reliable capture methodologies. Each method has its advantages and disadvantages, and the choice of method depends on the sculpture in question [9].

When feasible, computed tomography (CT) could be a valuable examination technique for 3D visualization and analysis of the internal structure of wooden sculptures (life size), their construction with various joint materials (nails, dowels of different wood type) or internal damages (cracks, woodworm tunnels) [10].

New technologies are becoming increasingly important in the field of conservation for digital records of artworks for research, documentation, reproduction, and treatment purposes. In certain cases, they can help to reassemble (broken) sculptures [11], or they can be used for the preparation, virtual sculpting, and printing of missing elements of artworks [12]. Three-dimensional data of artworks also help in the production of storage and transport packaging for museum collections [13].

Their use for documentation purposes is probably even more important. They allow for 3D visualization of most of the details of the sculpture, which are not only viewed visually, but the 3D models could also be "enriched" with the data obtained from material analysis (FTIR, micro RAMAN). Such an interactive presentation of the object could serve as a reference for recording and depicting areas of decay, repair, intervention, and diagnostic intervention as well as a starting point for further research [14].

Digital approaches to the treatment of cultural heritage are marked by differences compared to traditional approaches. Although the methods are often combined in practice, the importance of digitization and its collaboration with machine learning approaches in documentation, presentation, and interpretation is steadily increasing. Regardless of the type of approach, the goal in interpretation remains to increase public accessibility to the heritage object, improve audience understanding, increase participant interest in the experience, and pleasure and enjoyment [15]. As Knudson states, the goal of interpretation is "the development of an informed and experienced citizenry in relation to our cultural heritage", which requires the interdisciplinary collaboration of experts from various fields (landscape architects, archaeologists, historians, designers, UX planners, psychologists, anthropologists, etc.) in implementing interpretive approaches.

Interpretive approaches facilitate both emotional and cognitive engagement of participants with cultural heritage and increased memorability and learning efficiency. Targeted

interpretive approaches also allow for universality of heritage presentation beyond cultural, sociological, religious, and gender issues. In addition, interpretive approaches contribute to increased accessibility of cultural heritage, which is consistent with sustainability goals [16].

Interpretive approaches are also responsible for cultural heritage being increasingly presented in an educational context, even outside of research in archaeology, history, anthropology, cultural studies, computer science, graphic design, didactics, sociology, natural sciences, etc. At both formal and informal educational levels, interpretive approaches are being skilfully used to integrate cultural heritage into presentation content and learning materials [17].

The presented research addresses the collection of Forma Viva's collection, wooden sculptures displayed outdoors in a natural environment near the Božidar Jakac Art Museum in Kostanjevica na Krki in Slovenia.

The conservation of outdoor wooden sculptures is a complex issue, where traditional conservation approaches are limited. Large sculptures that are in a poor state of preservation require even more technically demanding, laborious and often financially costly conservation and restoration measures. Although they do not replace physical preservation, new technologies could be beneficial in many ways. For one, they could expand the conservator's ability to examine, document, and assess condition. Digitization also allows us to preserve at least the cultural and historical values of those art objects that cannot be preserved for various reasons. It can be used as a tool to promote cultural, social, and economic values, but in doing so, we must respect and protect intellectual property and work in an interdisciplinary manner [18].

## 2. Aim of the Research

The goal of the research was to apply digital technologies to selected sculptures from Forma Viva's collection. We have developed an interdisciplinary approach to experiment with new ways of graphic documentation on a 3D model, a visualization that allows for a clearer representation of the examination of the original structural identity of the sculpture (technological part: carving, construction, joinery), but also to assess the extent of deterioration processes and consequently to assess the need for intervention (treatment) requirements. The method has been shown to be useful for documentation purposes in the past [14,19].

Experimental work has included digital documentation, digital 3D presentation, and the introduction of 3D interpretive approaches to wooden sculptures.

The research is presented through case studies of two wood sculptures from the above-mentioned collection. The two sculptures were created almost 40 years apart, which indicates the different states of preservation of the two statues. As a result, the treatment of these sculptures also offers different approaches to the treatment and implementation of 3D technologies. The first case is Jon Oxman's Tower of Babel, wood, 480 × 60 × 55 cm (1978), which has been studied, documented, analysed, and evaluated through visual observation, but also through digitization, photogrammetry, schematic 3D representation of damage, interactive 3D presentation, 3D printing, and interpreted into useful objects and interpretive presentations in audio–visual media (jewellery, animation). The Tower of Babel sculpture was created in 1978. The sculptor Jon Oxman was born in the United States of America in 1955. He graduated in sculpture and improved his drawing skills, and he studied ceramics. As for the origin of the sculpture, we know that the inspiration was a well-known myth, which the sculptor interpreted through the placement of wooden segments and the red colouring in the areas where the wooden segments move away from the vertical structure. The second example is Ryszard Litwiniuk's, Transition, wood, 350 × 350 × 296 cm (2017), which was modelled, textured, and visualized in 3D. Ryszard Litwiniuk is a Polish sculptor born in 1966 who works in the fields of sculpture, graphics and installation. The basic materials of sculpture are wood and stone, which, along with metal, are used in the Transition statue of 2017. The "Transition" sculpture has a constructivist structure, in the form of a circle made up of individual segments. This raises the question of time, of cyclical

paths of progress and thus of constant growth. The visitor can experience it as a door and thus as a "passage of time in space". Digital presentations of the two statues were prepared for exhibition purposes and as useful objects (jewellery) with interpretation and storytelling approaches.

Transferring the idea of a wooden sculpture materialised by the artists into a new interpretation through different levels of digital representation is also of great importance from the point of view of storytelling.

## 3. Experimental Part

The workflow for 3D presentation and interpretation of wooden sculptures was divided into three approaches: a conservation approach, a technological approach, and a design–creative approach, as shown in Figure 1.

| WOODEN SCULPTURES FORMA VIVA | | DIGITIZATION | IMAGING / REPRESENTATION | 3D PRINTING |
|---|---|---|---|---|
| **CONSERVATION APPROACH** documentation and detail describing of sculptures | | SKETCHING measuring | 2D / 3D VIZUALIZATION OF DAMAGE (teksturing with new patterns on 2D / 3D models ) | |
| **TECHNOLOGICAL APPROACH** production of "authentic„ digital and printed reproductions of sculptures | | 3D MODELING (accurate)  PHOTOGRAMETRY - dron - digital camera | VIZUALIZATION (teksturing)  retopology | SCULPTURAL PROTOTYPES post-processing  Materials: ▪ thermoplastics ▪ photopolymers ▪ metals |
| **DESIGN APPROACH** creative interpretation of sculptures | | 3D MODELING (creative)  conceptual studies visualization on the body | DIGITAL ANIMATION (skulpture / narration)  various animation techniques | JEWELERY post-processing |

**Figure 1.** Schematic presentation of workflow of digital preservation, 3D reproduction and interpretation of Forma Viva sculptures from Kostanjevica na Krki.

The properties of the sculptures considered for the 3D presentation were divided into the following categories:

1. Damage of the sculptures (major damage: preservation—or still together parts of the sculpture, missing parts of the sculpture and minor damage: cracks, insect damage, lichen and moss, etc.);
2. Material of the sculptures (wood, possible additions of stone, metal, etc.),
3. Surface treatments/layers (coatings, paints, etc.),
4. Size and location in the environment.
5. Morphological properties of the sculptures (number of details, orientation of the surfaces of the sculptures, geometric orientation and organicity, complexity and branching).
6. Suitability of the sculpture for the photogrammetry or for the 3D modelling process.

The selection of sculptures for analysis and 3D presentation was based on the characteristics of the artworks for conducting the procedure, which were determined during the analysis of the sculptures in the field. Of all (more than 100, but today around 80 remain-

ing) the outdoor sculptures in the collection, 24 sculptures were selected for capture with photogrammetry, and 19 sculptures were selected for 3D modelling.

The methodology presented below for the two selected statues was determined on the basis of multiple attempts at the appropriateness of different approaches in research, art and didactic (collaboration with students) terms, some of which were also abandoned (e.g., 3D scanning, application development). The statues considered in this research represent two "extremes". A statue exhibited more than forty years ago, Tower of Babel, is heavily damaged, and its condition has changed significantly from its original form, requiring a photogrammetric approach to capture it, and Transition, exhibited recently, in which its good preservation state allowed only for documentation of the existing condition and the use of 3D modelling for 3D recreation.

The approaches to the sculptures thus represent the optimal framework from documentation to interpretation, which was determined for the selected sculpture based on two years of interdisciplinary research collaboration among the restoration, computer, and graphic arts professions.

### 3.1. Conservation Approach

We carefully assessed the condition of the sculpture, which is important for the development of the conservation and restoration plan [20]. At the same time, we collected archival records about the statues and their condition.

The sculptures were documented using traditional conservation in situ sketches (simplified linear drawings), measurements, photographs, video recordings, and written documentation, as well as the process of photogrammetry and 3D modelling, which also served for further procedures of 3D presentations and interpretations.

After the completion of the 2D graphic documentation of the Tower of Babel, our goal was to display/transmit the same information on the 3D model (see below).

### 3.2. Technological Approach

The goal of the technological approach was to obtain the most realistic digital reproductions of the original wooden sculptures. The first step was to digitize the sculptures. We used 3D modelling and photogrammetry to capture the data of the sculptures. Due to equipment (scanner) limitations and the size of the outdoor sculptures, 3D scanning was not possible.

We decided to use 3D modelling for sculptures that would be difficult to capture with 3D photogrammetry (holes, large overhangs) or whose morphology is simple (and therefore more suitable for modelling). The technological approach also included visualization with texturing, interactive presentation of the digitized sculptures, and 3D printing of the selected sculptures.

### 3.3. Photogrammetry

Photo recording was carried out with UAV (Unmanned Aerial Vehicle), i.e., DJI Mavic Pro Platinum (SZ DJI Technology Co. Ltd., Shenzhen DJI Sciences, Technologies Ltd., Nanshan, Shenzhen, China), equipped with a standard 3-channel RGB camera, and handheld digital cameras (Sony alpha a6000 and alpha 7 mk I, Sony, Minato, Tokyo, Japan) for capturing the parts of the sculptures that were inaccessible for drone recording. For certain wooden sculptures (exhibited outdoors), the placement of the sculpture (proximity to other objects such as trees, walls, etc.) also makes it necessary to combine (image) captures from different devices. The correct import and coordination of the photographs in photogrammetry programs was required further on (for the program to treat them correctly in terms of focal length, lens, angle, spherical aberrations). The number of drone photographs averaged 380, and we used between 50 and 150 additional photographs, depending on the sculpture. Photogrammetry was performed in Agisoft Metashape (Agisoft LLC, Sankt Petersburg, Russian Federation). Some sculptures required mesh correction, model, and texture optimization for real-time web presentation after the polygonal mesh was created

(missing parts of the sculptures). This was carried out in Blender (Blender Foundation, Amsterdam, Netherlands).

The Tower of Babel was photographed three times from 209 to 350 photos and photogrammetrically processed in Meshroom (AliceVision framework, Paris, France) and Agisoft (Metashape). The photogrammetry process in Agisoft included the steps Align photos, Build dense cloud, Build mesh and Build texture. In Meshroom, the process consisted of CameraInit, Depth Map, Depth Map Filter, Feature Ex-traction, Feature Matching, Image Matching, Mesh Filtering, Meshing, Prepare Dense Scene, Structure from Motion, and Texturing. Figure 2c shows the texture of MeshRoom and Agisoft. The results in both cases were satisfactory for further work; however, we used models and textures from Agisoft. Figure 2d shows the presentation of the sculpture on the Sketchfab web 3D viewer.

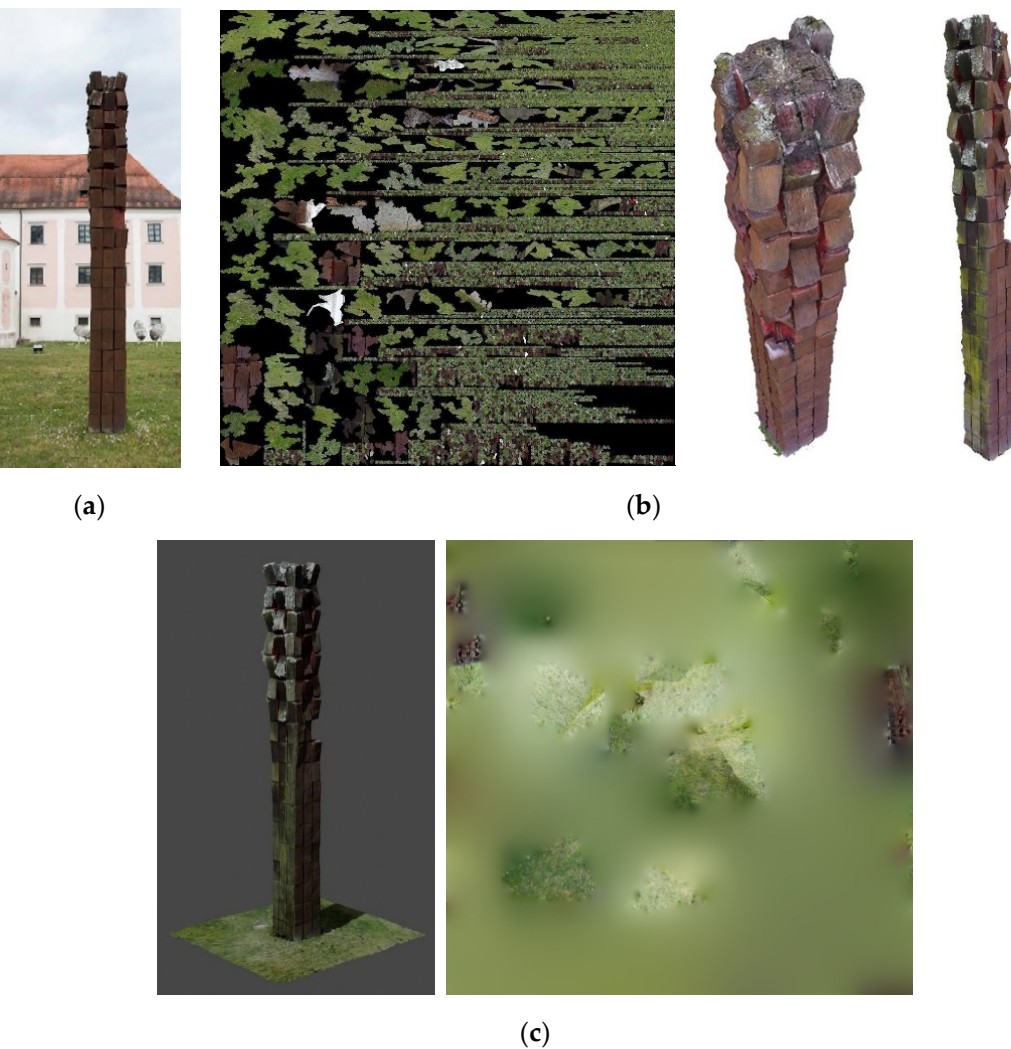

(**a**)　　　　　　　　　　　　　　　　　　(**b**)

(**c**)

**Figure 2.** *Cont.*

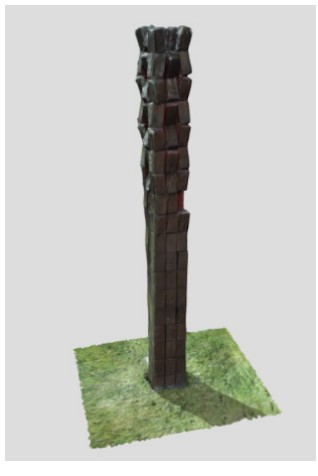

(**d**)

**Figure 2.** Different stages of representation. (**a**) Tower of Babel (Jon Oxman, 1978); (**b**) texture and photogrammetric 3D presentation in MeshRoom (Authors of 3D presentation: Luka Obal, Alja Berlot); (**c**) texture and photogrammetric 3D presentation in Agisoft Metashape (authors of 3D presentation Urška Klenovšek, Karmen Kogej, Hana Ocvirk, Mylana Peresypkina); (**d**) representation of sculpture on Sketchfab.

### 3.4. 3D Graphic Documentation

The method was used to first create a high polygon 3D model and then gradually simplify it to reduce the number of polygons (up to 500,000 polygons/faces) to preserve as much detail as possible while increasing the speed of processing, and secondly, to export the 3D model without the unnecessary environment of the sculpture, which allowed us to extract the texture with only essential components of the object.

In order to create a graphical documentation, the patterns had to be defined beforehand (Figure 3). The goal was to generalize the patterns for the conservation community. A collection of textures representing different types of damage and other conditions was created. The patterns for the hatches were chosen carefully. The main concern was to select textures that matched each other so that they could be overlayed. They were then colour coded, e.g., textures coloured green represents vegetation while red illustrates damage to the support. A translucent background was added to some of these textures so that they can be seen even when the applied area is small.

The texture of the 3D model was edited with Adobe Photoshop® (Adobe complex, San Jose, CA, USA) image processing software. It was used as a background, where hatches of all sculpture's conditions were applied to a newly created layer (Figure 4), which was saved as a new texture in the same format as the original one. In order to use the resulting 3D model in presentation software for presentation of 3D models, it was exported to a .3mf file, which was then combined with the texture (Figures 5 and 6) in the 3D Builder program (Microsoft, Redmond, WA, USA).

The 3D web viewer Sketchfab (Sketchfab, Paris, France, New York City, NY, USA) was used for the interactive 3D presentation of the virtual models (https://sketchfab.com/reformaviva, accessed on 8 June 2022). Figure 7 shows the 3D presentation of sculptures in Sketchfab.

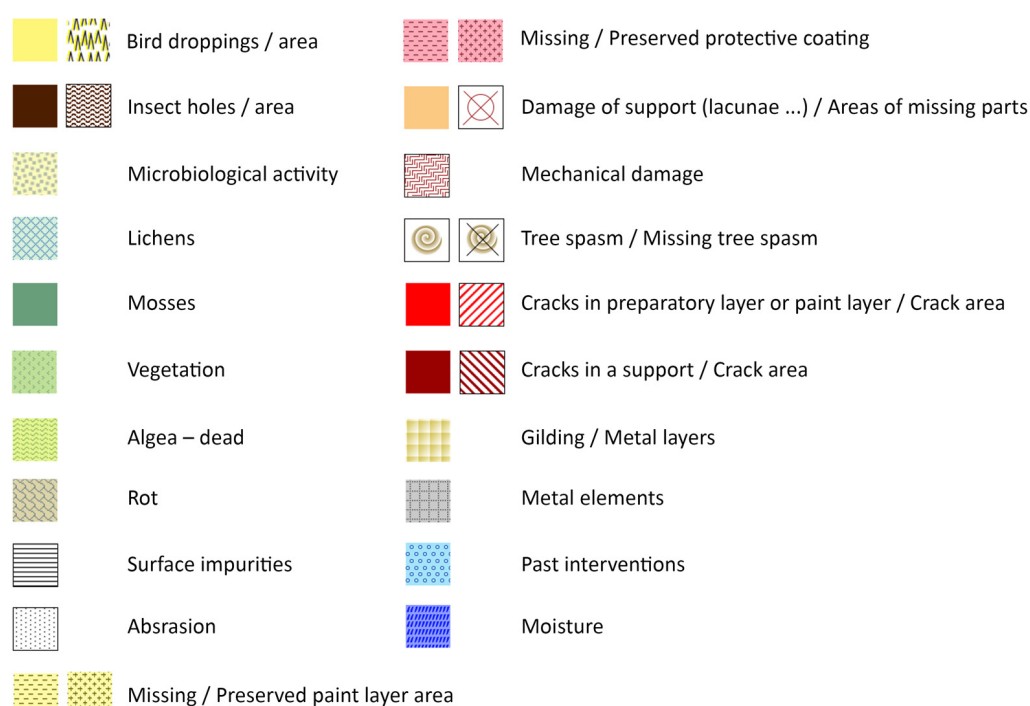

**Figure 3.** Legend of patterns representing different types of damage and other conditions (authors: Katarina Bartolj, Eva Marija Fras, Marko Odić, Lara Skukan, Ana Sterle, Sara Štorgel).

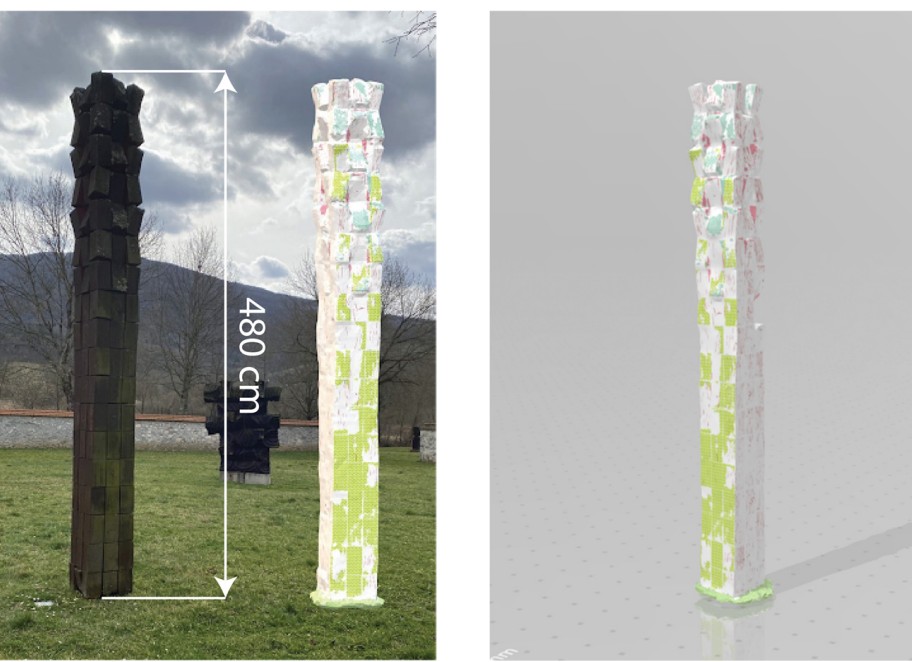

**Figure 4.** A result of the 3D graphic presentation of the sculpture *Tower of Babel* pasted next to the original sculpture (mentor: Prof. Dr. Denis Trček; Author: Lara Skukan).

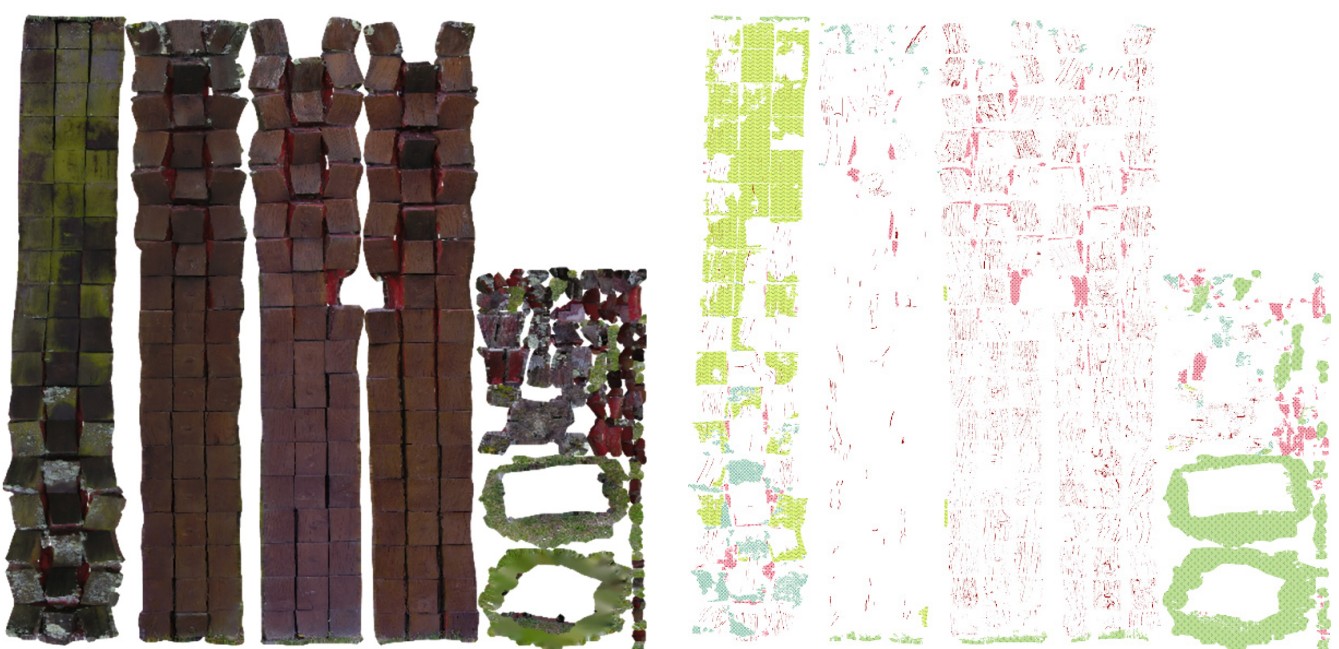

**Figure 5.** The texture before and after the applied hatches (mentor: prof. dr. Denis Trček; author: Ana Sterle).

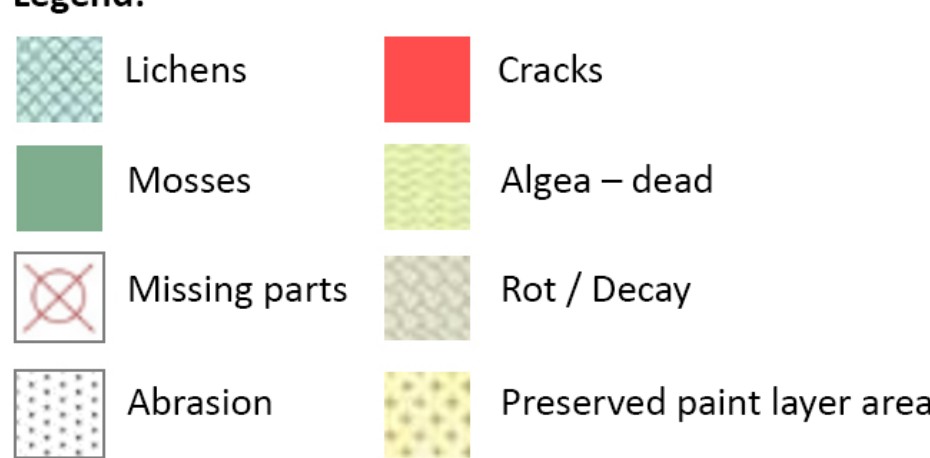

**Figure 6.** Legend of applied hatch patterns used on Figures 4 and 5 (mentor: Prof. Mag. Tamara Trček Pečak; authors: Lara Skukan, Ana Sterle).

The results of 3D graphic documentation and mapping of graphic patterns showing damage and other conditions are new to the field of documentation and conservation interventions. The usefulness for research can be seen in the further study and investigation of sculptures, especially with the ability to 3D track the condition of the sculpture over time.

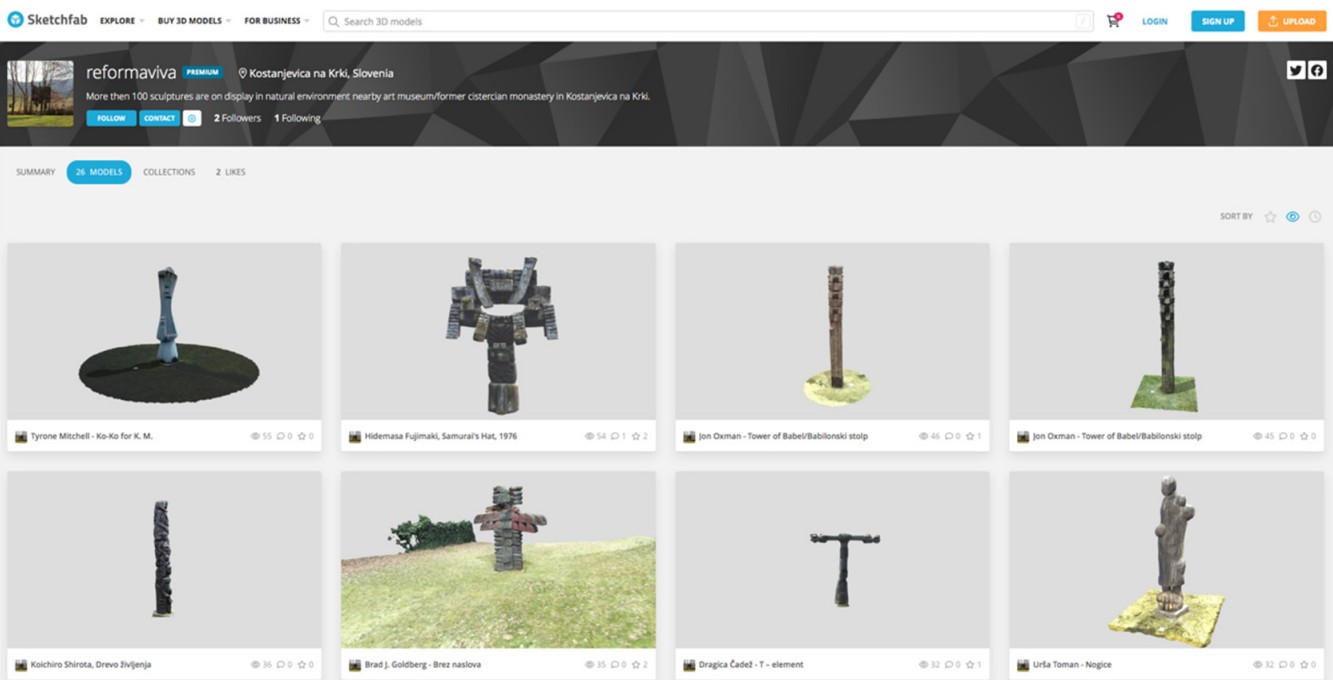

**Figure 7.** Three-dimensional presentations of sculptures in 3D viewer Sketchfab.

### 3.5. 3D Modelling

The process of 3D modelling was applied to sculptures without many organic shapes, to sculptures with many open parts and to those that are geometrically more regular. Such sculptures can be easily described with ordinary Euclidean geometry.

Of the two sculptures we used for the study, the Transition sculpture was better suited for 3D modelling.

The first requirement for modelling was to have the exact dimensions of the parts of the sculpture. Figure 8 shows a sketch with the dimensions of each part.

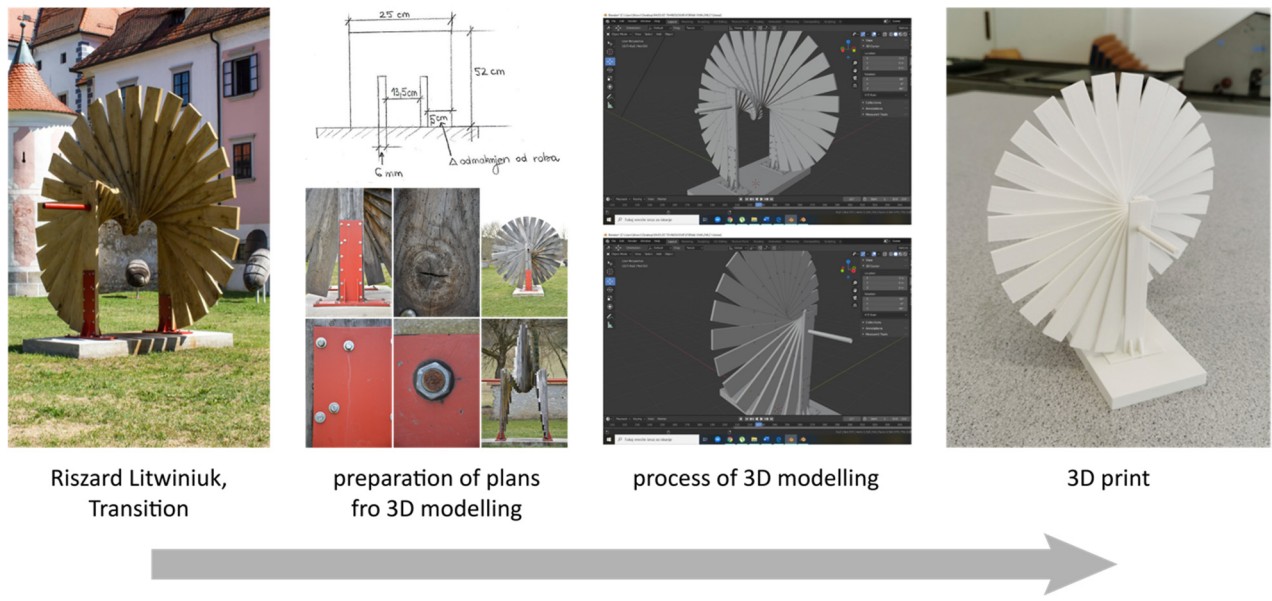

Riszard Litwiniuk, Transition    preparation of plans fro 3D modelling    process of 3D modelling    3D print

**Figure 8.** The process of the technological approach to the reconstruction of the sculpture "Transition", Ryszard Litwiniuk, 2017 by the process of 3D modelling (authors: Eva Bužanin, Monika Samsa, Nejc Suhadolnik).

In addition to accurate sketches, modelling also requires high-quality photographs of the entire sculpture, its details, and textures (Figure 8). The modelling of the Transition sculpture was simple. Mainly "primitive" mesh shapes such as planes, cubes and cylinders were used for modelling. First, a concrete base was modelled on the base of the cube, and then a board and a beam. If the sculpture has two beams, another one was created using the mirror transformer. The board was placed over the beam, and then, the tube was added with a solidify transformer to form a tube. An empty object was added to the centre of the tube to serve as a reference point for the array transformer. Additional boards (27 of them) were created using the transformer mentioned above. Each board was rotated clockwise around the centre of the axis at an angle of 13.3° to the previous one.

The steps of modelling parts of the Transition sculpture, i.e., the lower support part and the main elements of the sculpture, are shown in Figure 8. The exact modelling, including the screws, was performed only for the final visualization with texturing. The final visualization of the sculpture can be seen in Figure 9.

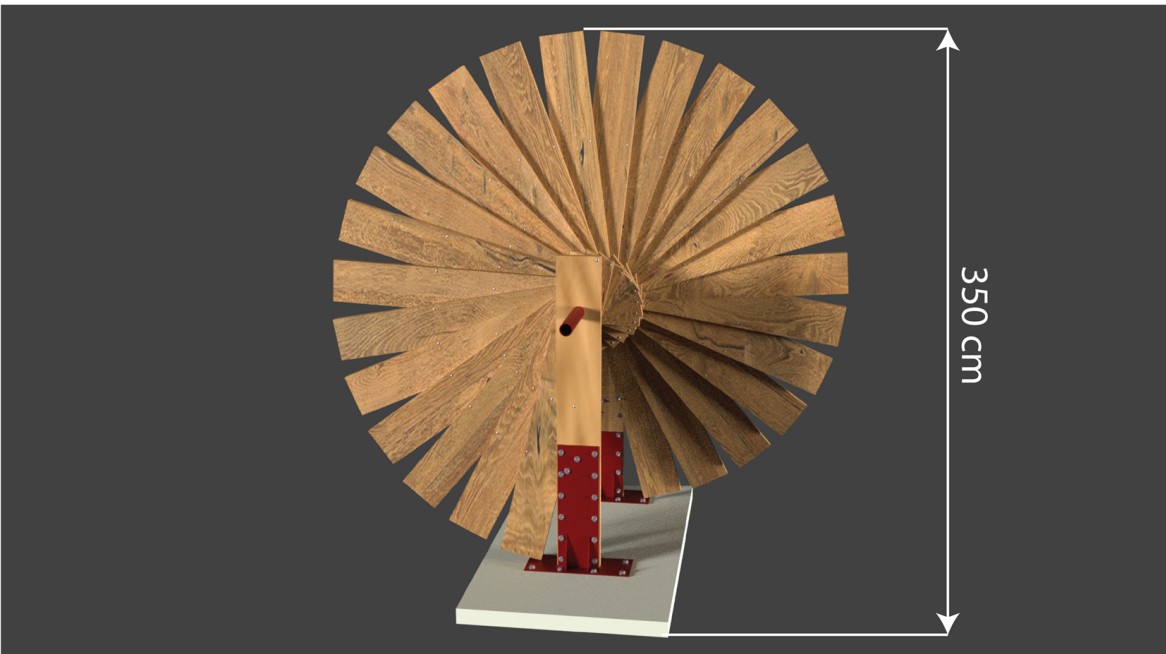

**Figure 9.** Final visualisation of the sculpture Transition, Ryszard Litwiniuk 2017 (authors of 3D model and visualization: Gaja Hanuna, Simon Perovnik, Nina Krpan).

### 3.6. Model Preparation and 3D Printing

A ZMorph printer (ZMorph S.A., Wrocław, Poland) with a working volume of 235 (X): 250 (Y): 165 (Z) mm was used to print the prototypes of the wooden sculptures. It is based on the fused deposition modelling technology. The prototypes of the sculptures were printed using white thermoplastic PLA filament with a diameter of 1.75 mm. Both models presented in this paper were printed in 1:20 scale. We produced or printed all other sculptures included in the project research at the stated ratio. The exception was the sculptures that exceeded the printer's working volume despite the consideration of the 1:20 scale. There were three such sculptures among all sculptures selected for 3D printing that we had to further reduce in size.

Both prototypes of the sculptures (those created by photogrammetry and those created by 3D modelling) were printed using the same procedure. First, the meshes of the model were analysed in Blender (e.g., overhang, thickness, intersections, distortions, sharp edges). Then, the optimal orientation of each model was determined. The model had to be oriented to achieve the best print quality. In addition to high visual quality, model orientation affects mechanical properties, optimal printing time, and optimal material consumption (support

structures, fill density, number of walls). An appropriately prepared print model was exported in the form of an .stl file.

The basic printing parameters were chosen to achieve good print quality and rational energy consumption (printing time and material consumption).

The layer thickness was set at 0.2 mm. The filling used was a honeycomb form with 30% density of the inner filling. The printing speed was set to 30 mm/s. In addition to the filling pattern, three outer walls were used for printing each layer.

Voxelizer software was used for the final preparation of the models and creation of the G-code. The images of the model preparation of the transition sculpture for printing and the final prints can be seen below in Figure 10. Figure 11 shows the result, the printed and assembled scale model of the Transition sculpture and the 3D printed parts of the sculpture and the process of assembly.

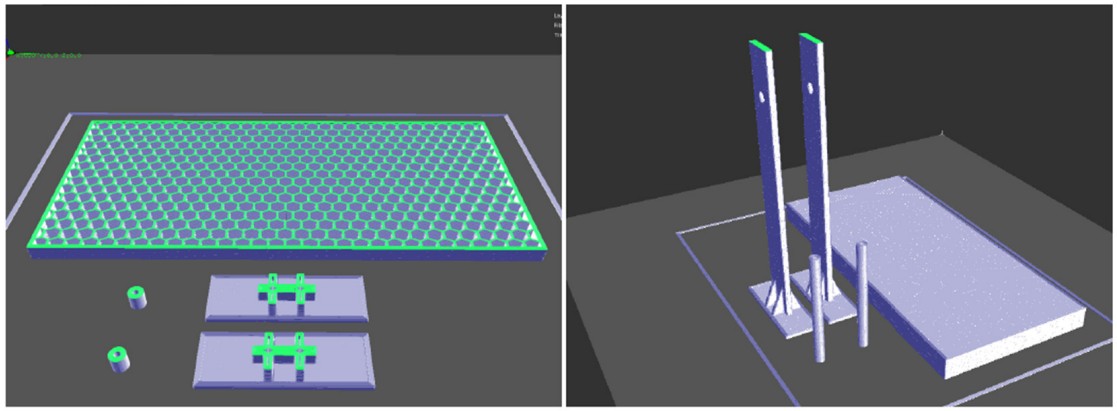

**Figure 10.** Three-dimensional printed model of the Transition sculpture.

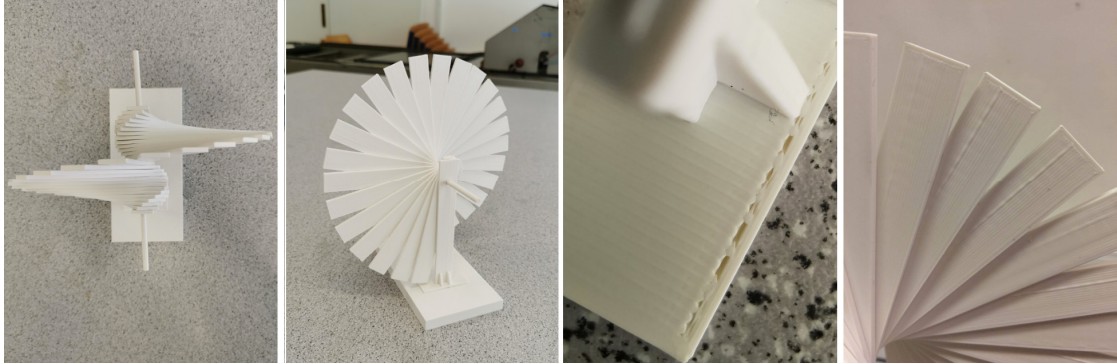

**Figure 11.** Three-dimensional printed parts of the sculpture Transition and the process of its assembly.

The applicability of the scope of technological approaches is a methodological innovation in the case of wooden cultural heritage. It included the integration of photogrammetry and/or 3D modelling and retopology, in the case of a final digital presentation of texturing and rendering, and in the case of a physical presentation the preparation for 3D printing and 3D printing, which with further interpretations increased the possibilities of using the products beyond research circles to the general public.

## 4. Interpretation of Digitized Sculptures

### 4.1. Interpretation in Jewellery

The goal of translating a sculpture into a jewellery collection was to understand the monument and translate its concept into a design piece made of a different material, but with the same purpose: to communicate with the viewer or user. In addition to

creating miniature replicas of works of art, it was also interesting to create new creative products that art galleries could offer as part of the sales programme. In this way, we have reinterpreted one artistic field—sculpture—into another artistic field—original jewellery. The new technologies of 3D printing from new materials, especially metal alloys, are ideal for the production of jewellery, whether one-offs or micro-series, due to their precision and material quality.

The design process began with the study of the two selected sculptures, which were analysed from visual, conceptual, artistic and production points of view, to then explore their interpretation in a jewellery collection.

In addition, the study of contemporary jewellery design was important. Therefore, we focused on conceptual jewellery design, where the value lies in the thought process or concept of a piece rather than the materials or forms used. For conceptual jewellery designers, the intent of a piece of jewellery is to provoke and stimulate reactions in the wearers and viewers.

The design process begins with initial sketches of design ideas. The shape and size of the jewellery must consider the shape of the human body and be adjusted as necessary.

The process continues with 3D modelling of the jewellery using the Blender program. Once the detailed 3D models of the jewellery were designed, they had to be prepared for 3D printing by defining their exact dimensions and preparing them with the chosen technology. The 3D models of the jewellery created in Blender are shown in Figure 12. The weight of the jewellery also had to be considered.

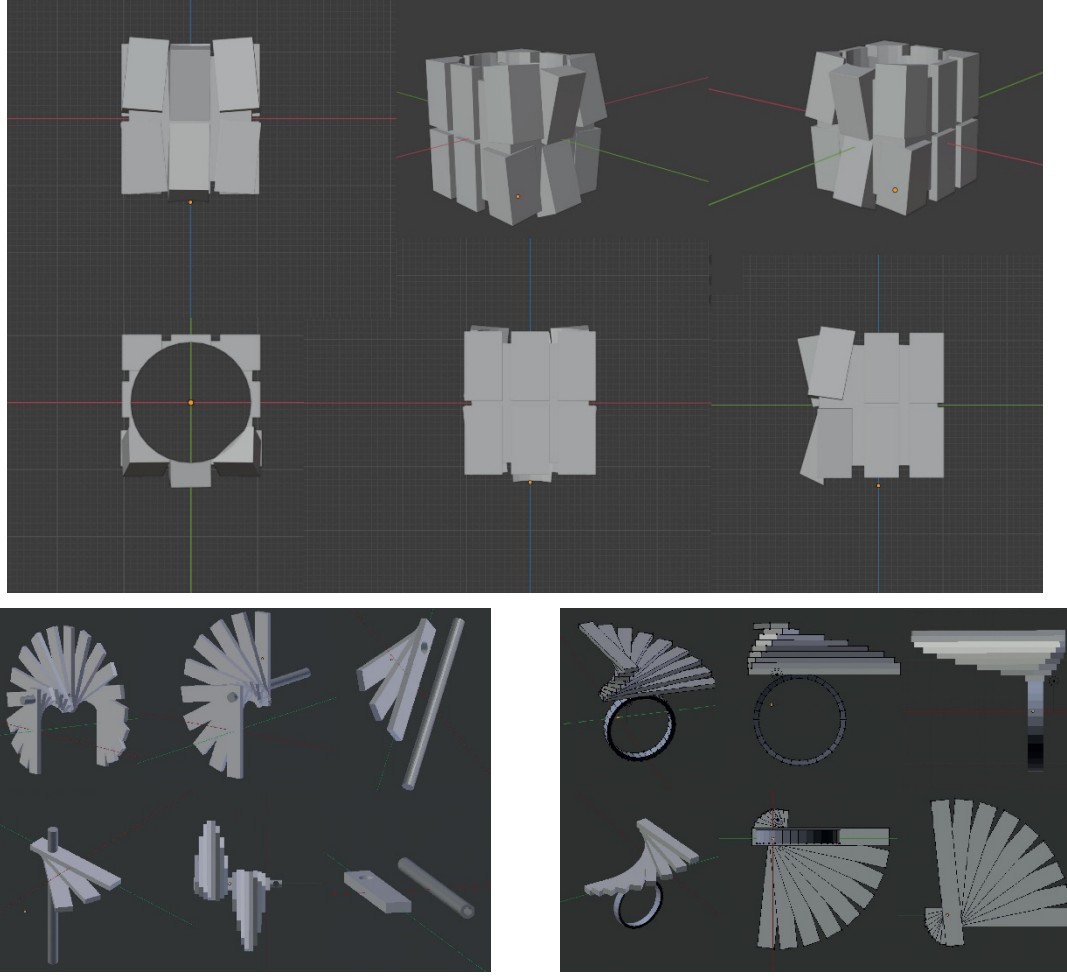

**Figure 12.** Three-dimensional models of jewellery created in Blender.

Two different printing technologies have been used for 3D printing: Fused Deposition Modelling (FDM) technology and Selective Laser Melting (SLM). The latter technology was used to print the jewellery with metal powder. The jewellery, based on the Transition sculpture, was printed using FDM technology with a polymer containing 40% recycled wood (Figure 13). The texture is rougher, and the colour is reminiscent of light wood. When printing with this polymer, the choice of the angle of layer deposition is important to create differences in the gloss of the surface of the PLA filament. The perimeter of the filament used was 1.75 mm, and the layer thickness was 100 microns. The infill pattern was linear in the horizontal direction with an infill density of 40%. Each layer was printed with three outer walls. SLM technology with metal powder was used to print jewellery inspired by the Tower of Babel sculpture. This technology allows for the production of jewellery with final functionality (Figure 14). For this purpose, a 100 W laser metal printer, LPM100 (Dentas, Maribor, Slovenija), and a metal powder alloy Co-Cr-Mo-W were used. The layer thickness was 100 micrometres. After printing with metal powder, the surface of the object feels rough. Therefore, we sandblasted and, in some cases, polished the jewellery.

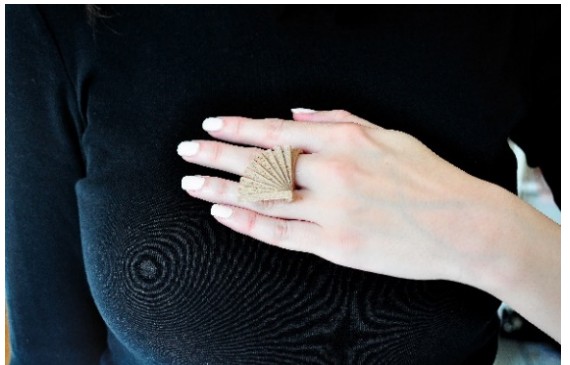
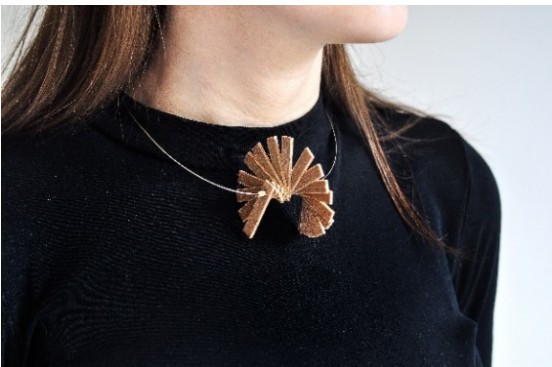
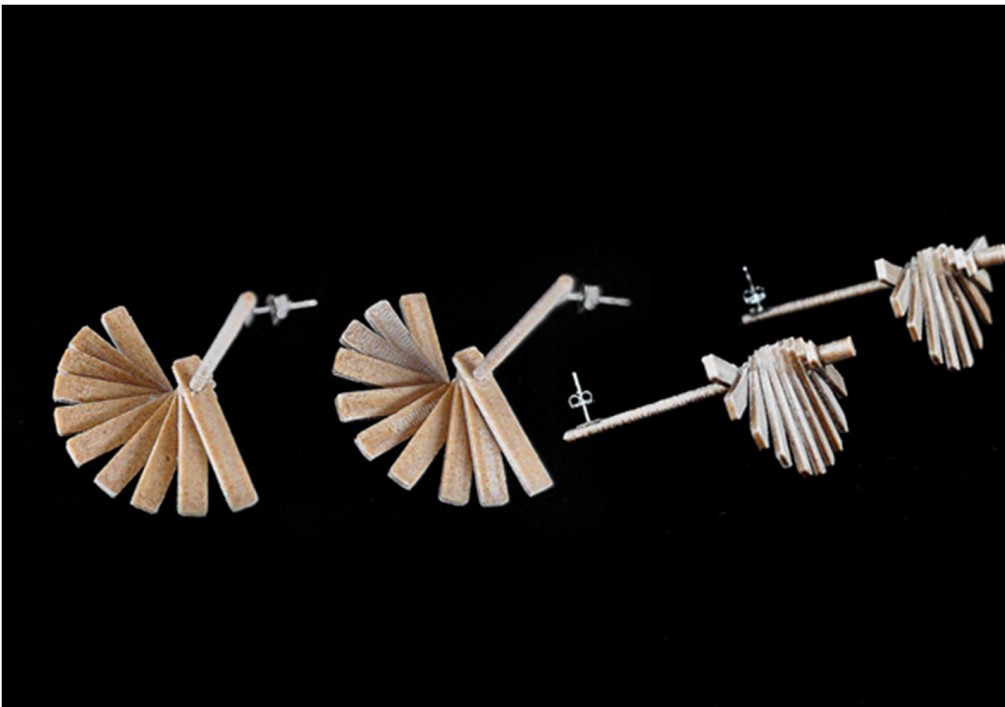

**Figure 13.** Collection of jewellery printed with polymer containing recycled wood (author: Katrin Večerina).

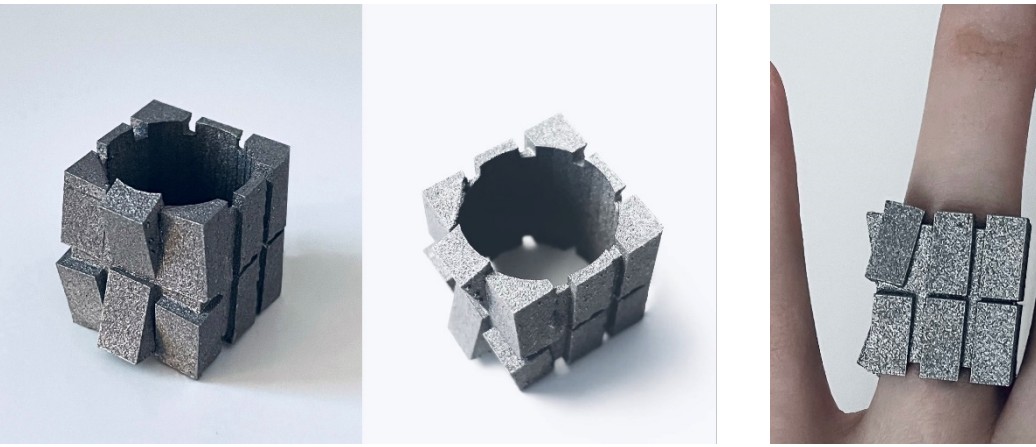

**Figure 14.** A ring printed with metal powder (author: Kaja Bakan).

### 4.2. Interpretative Animation

The interpretation process also included the production of 3D animations by placing digital sculptures in the story (following a short script proposal) and allowing users to experience the medium by presenting the sculpture in a more interactive and engaging way. These artistic interpretations promote and stimulate different perspectives on the sculptures of Forma viva and enrich the experience of cultural heritage in the Park of Sculptures in Kostanjevica na Krki. As an interpretive animation, we present the example of the Tower of Babel. The animation depicts the original story of the Tower of Babel, placing the sculpture in the desert landscape of the Middle East and explaining the origins of the diversity of languages and human attempts to reach heaven on Earth, with the addition of dramatic metallic materials, bright red textures, and a dynamic camera circling around the sculpture, highlighting the confusion, disappointment and defeat. After a dramatic display of confusion, the story and its interpretation are calmed by the presentation of a scene in which the sculpture is placed in a natural environment, bringing peace and knowledge about the tower myth and its meaning. Figure 15 illustrates the processes of technological and interpretive approaches to the *Tower of Babel*.

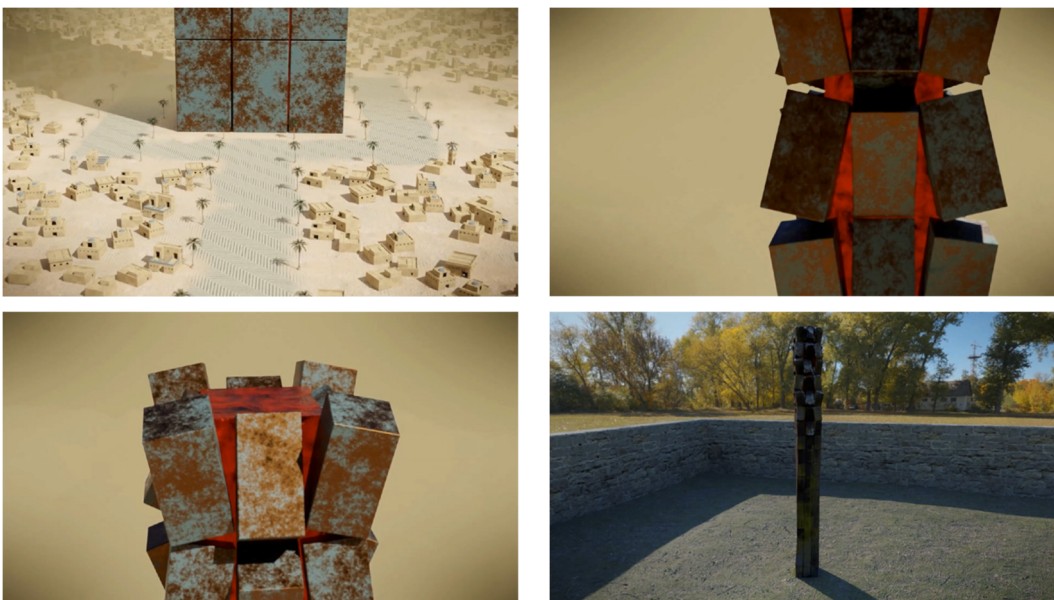

**Figure 15.** The process of an interpretative approach for creation of 3D animation of the sculpture "Tower of Babel", Jon Oxman, 1978 (author of 3D animation: Jernej Kalin).

Interpretive approaches also bring artistic and educational usability to the results. The possibility of interacting with sculptures is thus extended by researchers to users and to the realm of user experience, the introduction of multisensory (seeing, touching, hearing) digital and physical presentation media, and thus greater user engagement.

## 5. Conclusions

The paper presents an interdisciplinary approach to the treatment of the FormaViva collection of wooden sculptures exhibited by the Božidar Jakac Gallery (Kostanjevica na Krki, Slovenia).

The new technologies are not able to replace real artworks/sculptures, but they can significantly improve the way conservators examine, document, and assess their condition. Various 3D methods were applied to selected sculptures, which informed the documentation process that serves as the basis of planning preservation. Three-dimensional representations provide more comprehensive records of the physical structure and condition of the artworks at a given point in time (evidence of their history), which is needed prior to any intervention. Three-dimensional graphic presentations allow conservators to visualise certain data that are difficult to achieve with (traditional) 2D presentations, but their use requires new skills, knowledge, and additional equipment or the necessary collaboration with other experts, which is not always possible.

Digitization also allows us to preserve at least information about those art objects that cannot be physically preserved for various reasons. High-quality data can be made available (digitally) to experts for research or to the public worldwide to promote artworks in various ways, respecting an artist's intellectual property.

Interpretive approaches, user-centered design approaches, and storytelling principles have been applied to 3D representations of statues created with photogrammetry and 3D modelling. The results are photorealistic renderings, interactive presentations (SketchFab 3D viewer), 3D-printed reproductions, jewellery, and interpretive animations. These approaches open new ways of presenting the heritage of the wooden statues and, in particular, their digital conservation, which expands the possibilities for disseminating the research findings in the public space.

The results of the research show that the procedures of graphic documentation on 3D models, innovatively introduced in 3D form on wooden heritage objects in this research, allow for a more detailed investigation of the original structural identity of the sculpture. This is especially important when artworks are in situ and post-processing the 3D object with two-dimensional recorded information becomes difficult. By analysing 3D graphic documents and incorporating 3D and interactive technologies (3D printing, interactive 3D presentations), we expand the usability of cultural heritage objects and create new modalities of use (tactility, interactivity). By using interpretive techniques that resulted in jewellery and interpretive animations in our research, we not only breathe new life into the sculptures, but also enrich the stories of the sculptures with our own experiences of the sculptural work and the form and expressiveness of the sculptures.

The story of the sculptures was told in an innovative way that has a great impact on the understanding and personification of the work of art.

**Author Contributions:** Conceptualization, D.T., K.K., H.G.T. and T.N.K.; methodology, A.U. and D.M.; software, A.U. and A.S.; validation, M.V., H.G.T. and T.N.K.; investigation, T.T.P. and D.T.; resources, M.V. and D.M.; writing—original draft preparation, A.S., M.V., K.K., D.M., H.G.T. and T.N.K.; writing—review and editing, J.A., H.G.T. and T.N.K.; visualization, J.A., H.G.T. and T.N.K.; supervision, T.T.P., H.G.T. and T.N.K.; funding acquisition, H.G.T. All authors have read and agreed to the published version of the manuscript.

**Funding:** This research was co-funded by the European Union's Erasmus+ program, KA203—Strategic Partnerships for Higher Education and by the Slovenian Research Agency (Program P2-0213 Textiles and Ecology).

**Institutional Review Board Statement:** Not applicable.

**Informed Consent Statement:** Not applicable.

**Data Availability Statement:** Not applicable.

**Acknowledgments:** We would like to thank our colleagues and project partners from Božidar Jakac Art Museum, Estonian Academy of Arts (Department of Cultural Heritage and Conservation), and Academy of Fine Arts Zagreb (Department of Art Conservation and Restoration), who greatly helped us with their insights and expertise in conducting the research. We would like to thank all the first and second level students of Graphic and Media Technology, Graphic and Interactive Communication, and Textile and Fashion Design, with whom we collaborated with during the three years of the project in the subjects of Digital Design, Basics of 3D Modelling, and Advanced 3D Computer Graphics and Visualizations, and whose work contributed to the excellent results of the project. Working with them was the essence of the project.

**Conflicts of Interest:** The authors declare no conflict of interest.

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
