# Peer review of "3D Digital Preservation, Presentation, and Interpretation of Wooden Cultural Heritage on the Example of Sculptures of the FormaViva Kostanjevica Na Krki Collection"

_applsci, doi:10.3390/app12178445_

Round 1

Reviewer 1 Report

The work looks as good student presentation. Perhaps it would be interesting to add a historical and cultural reference about the objects under consideration.

It seems for me, that there are a superfluous number of photos at figures 10 and 11

Reviewer 2 Report

This innovative and interesting research describes the reproduction of two artefacts using a variety of technical methods.

The 'state of the art' can be found in the Introduction. The sources used are good.

The textual inconsistency between paragraphs could be improved. The research and artistic usability of the virtual and material content created in the project should be explained precisely and concretely here. 

In the experiments presented on the two artefacts, the technical operation should be described precisely and concretely, the exact purpose of the research (purpose of the choice of different techniques - comparison, integrative method, etc.?)

In the Conclusion, the concrete research objective should be evaluated.

'Experimental work has included digital documentation, digital, 3D presentation, and the introduction of 3D interpretive approaches to wooden sculptures.'

It is necessary to describe why these technical methods were chosen.

'Translating a sculpture into a jewellery' is a very interesting tool, but needs to explain why it was chosen.  

The Conclusion should reflect in more depth on the 'state of the art' described nicely in the Intorduction.

Reviewer 3 Report

The article publishes data for digital archived on collection of wooden sculptures. The publication methodically describes the sequence and possibilities of modern digitization tools. The possibilities of technical means are considered with the dimensions of the objects.

The paper can be considered for publication in this Journal provided that the authors can further address the following issues:

1. The font in Figure 3 that describes the textures is small and difficult to read when paper is printed;

2. In figures 4, 8, 9 and 11, it would be better to add scale lines to the different images, because the reader loses an idea of the actual size of the objects. Looking at the original photo of the sculpture "Transition" (Figure 8) and its printed copy. In front of me, the question arises, are their sizes the same?

3. In the text to point 3.6 it is described about the working volume of the 3D printer, the scale relative to the original objects, but it is not clear whether it applies to all objects or only to some.

4. The photograph with the dimensions of figure 8 is not readable even when the document is zoom in digital version. It is necessary either to enlarge the photo itself or to replace it with one drawn in some CAD system.
